# Chest Tube Placement in Mechanically Ventilated Trauma Patients: Differences between Computed Tomography-Based Indication and Clinical Decision

**DOI:** 10.3390/jcm11144043

**Published:** 2022-07-13

**Authors:** Manuel Florian Struck, Christian Kleber, Sebastian Ewens, Sebastian Ebel, Holger Kirsten, Sebastian Krämer, Stefan Schob, Georg Osterhoff, Felix Girrbach, Peter Hilbert-Carius, Benjamin Ondruschka, Gunther Hempel

**Affiliations:** 1Department of Anesthesiology and Intensive Care Medicine, University Hospital Leipzig, Liebigstr. 20, 04103 Leipzig, Germany; felixfrederic.girrbach@medizin.uni-leipzig.de (F.G.); gunther.hempel@medizin.uni-leipzig.de (G.H.); 2Division of Trauma Surgery, Department of Orthopedics, Trauma and Plastic Surgery, University Hospital Leipzig, Liebigstr. 20, 04103 Leipzig, Germany; christian.kleber@medizin.uni-leipzig.de (C.K.); georg.osterhoff@medizin.uni-leipzig.de (G.O.); 3Department of Diagnostic and Interventional Radiology, University Hospital Leipzig, Liebigstr. 20, 04103 Leipzig, Germany; sebastian.ewens@gmail.com (S.E.); sebastian.ebel@medizin.uni-leipzig.de (S.E.); 4Institute for Medical Informatics, Statistics and Epidemiology, Medical Faculty, University of Leipzig, Härtelstr. 16-18, 04107 Leipzig, Germany; holger.kirsten@imise.uni-leipzig.de; 5Division of Thoracic Surgery, Department of Visceral, Transplant, Thoracic and Vascular Surgery, University Hospital Leipzig, Liebigstr. 20, 04103 Leipzig, Germany; sebastian.kraemer@medizin.uni-leipzig.de; 6Department of Diagnostic and Interventional Radiology, University Hospital Halle, Ernst-Grube-Str. 40, 06120 Halle, Germany; stefan.schob@uk-halle.de; 7Department of Anesthesiology, Intensive Care and Emergency Medicine, Pain Therapy, Bergmannstrost Hospital, Merseburger Str. 165, 06112 Halle, Germany; peter.hilbert@bergmannstrost.de; 8Institute of Legal Medicine, University Medical Center Hamburg-Eppendorf, Butenfeld 34, 22529 Hamburg, Germany; b.ondruschka@uke.de

**Keywords:** chest trauma, pneumothorax, mechanical ventilation, chest tube placement, computed tomography

## Abstract

The rate of occult pneumothorax in intubated and mechanically ventilated trauma patients until initial computed tomography (CT) remains undetermined. The primary aims of this study were to analyze initial chest CTs with respect to the thoracic pathology of trauma, the clinical injury severity, and chest tube placement (CTP) before and after CT. In a single-center retrospective analysis of 616 intubated and mechanically ventilated adult patients admitted directly from the scene to the emergency department (ED), 224 underwent CTP (36%). Of these, 142 patients (62%) underwent CTP before CT, of which, 125 (88%) had significant chest injury on CT. Seventeen patients had minor or absent chest injuries, most of which were associated with transient or unrecognized tracheal tube malposition. After CT, CTP was performed in another 82 patients, of which, 56 (68.3%) had relevant pneumothorax and 26 had minor findings on CT. Sixty patients who had already undergone CTP before CT received another CTP after CT, of which, 15 (25%) had relevant pneumothorax and 45 (75%) had functionality issues or malposition requiring replacement. Nine patients showed small pneumothorax on CT, and did not undergo CTP (including four patients with CTP before CT). The physiological variables were unspecific, and the trauma scores were dependent on the CT findings for identifying patients at risk for CTP. In conclusion, the clinical decisions for CTP before CT are associated with relevant false-negative and false-positive cases. Clinical assessment and CT imaging, together, are important indicators for CTP decisions that cannot be achieved by using clinical assessment or CT alone.

## 1. Introduction

Severe chest injury (abbreviated injury severity score >3) is observed in 50% of all major trauma patients, and contributes to differences in intensive care therapy and mortality [1,2]. One of the most significant chest injuries is pneumothorax, which can ultimately lead to tension pneumothorax with clinical signs of shortness of breath, a high respiratory rate, chest pain, poor oxygen saturation, and attenuated breath sounds. Furthermore, subcutaneous emphysema, jugular venous congestion, tracheal shift, hypotension, and cardiopulmonary collapse may develop in due course [3]. The importance of immediate treatment by pleura decompression has been demonstrated in many studies, and is part of systematic treatment protocols and guideline recommendations [4,5,6,7,8]. An unrecognized or not appropriately decompressed tension pneumothorax may be associated with preventable deaths [9]. However, the correct diagnosis of a traumatic pneumothorax by clinical means may be challenging, especially in rescue situations. The diagnostic value of early total-body computed tomography (CT) scanning and its implementation in emergency algorithms has been one of the key achievements in trauma care in the past few decades, and contributes to a reliable and standardized detection of possible life-threatening pneumothorax [5,10].

In sedated patients under mechanical ventilation, the diagnosis of pneumothorax may be different from that for conscious patients, who may be able to complain of chest pain and present tachypneic. Furthermore, mechanical ventilation may be an iatrogenic risk factor for developing a tension pneumothorax due to the application of positive pressure and positive end-expiratory pressure (PEEP) [3,11]. The rate of occult pneumothorax in intubated and mechanically ventilated trauma patients until the initial CT remains undetermined. Moreover, studies focusing on risk factors and the clinical status of trauma patients with chest tube placement before and after CT are scarce.

The primary aims of this study were to analyze initial chest CTs with respect to the thoracic pathology, clinical injury severity, and chest tube placement before and after CT. The secondary aims were to analyze prehospital and trauma room physicians’ decisions to perform chest tube placement before CT, and the diagnostic capability of CT to predict patients needing chest tubes afterwards. A retrospective observational study was conducted at a single level-1 trauma center involving data of acute major trauma patients who underwent prehospital or trauma room intubation and mechanical ventilation prior to initial trauma-CT. Then, standardized parameters provided in the trauma scores in different categories of patients undergoing chest tube placement were presented and compared to address the aim of this study. 

## 2. Materials and Methods

The local trauma registry of the University of Leipzig Medical Center was reviewed for consecutive patients admitted between 01/2008 and 12/2019. The inclusion criteria were age ≥18 years; admission directly from the accident scene; emergency endotracheal intubation before the initial CT; an injury severity score (ISS); a thoracic trauma severity score (TTS) [12]; the Berlin polytrauma definition [13]; and the availability of data from initial trauma-CT imaging, including chest CT. The data were obtained from medical records, the radiological information system, and the digital picture archiving and communication system (MEDOS RIS version 9.3.3008, Nexus MagicWeb Version VA60C_0115, Visage Imaging, PACS: syngo.plaza, Siemens Healthcare, Erlangen, Germany). Sample size estimation was not performed due to the exploratory and retrospective study approach with mainly descriptive characteristics. 

### 2.1. General Management

The prehospital care of major trauma patients was performed by an emergency medical service (EMS) team led by an emergency response physician until hospital admission. Trauma room activation and management were organized according to the recommendations of the German Society of Trauma Surgery (DGU) [5]. The interdisciplinary trauma team performed a standardized assessment according to the advanced trauma life support (ATLS^®^) approach, in which all the procedures were performed by consultants or senior specialists [6]. Initial trauma-CT was the diagnostic standard, and was performed immediately after clinical assessment, whereas critically unstable patients could be transferred directly to the operating room or could undergo cranial CT only. Plain chest radiography was not performed in the acute emergency setting in any of the cases included. During the study period, there was no change in the clinical management of major trauma patients. In this study, we analyzed CT data for chest injury (lung contusions, rib fractures, pneumothorax, hemothorax) with regard to chest tube placement and clinical injury characteristics (e.g., PaO_2_/FiO_2_-ratio). Chest tube placement was analyzed regarding different time points (before and after CT). Chest tube replacement was performed after CT in the case of functionality issues and malposition. A pneumothorax was classified as a “clinically relevant” pneumothorax or “small pneumothorax” by two radiologists, a thoracic surgeon, and an anesthesiologist. 

### 2.2. Statistical Analysis

The data are reported as the mean (standard deviation, SD) for normally distributed variables, the median (interquartile range, IQR) for non-normally distributed variables, and as numbers (percentages), where appropriate. For comparisons, the Fisher’s exact test was used for qualitative data, and the Mann–Whitney U-test for quantitative data. The statistical evaluation followed a descriptive approach regarding the different groups for the location of chest tube placement: chest tube placement before CT (prehospital chest tube placement, trauma room chest tube placement, and a group who underwent chest tube placement prehospital and in the trauma room), and chest tube placement after CT (chest tube placement after CT and not before CT, chest tube placement before and after CT, and a subgroup of patients who received chest tube placement prehospital, in the trauma room, and after CT). The investigated variables for these group comparisons were age; sex; body mass index (BMI); PaO_2_/FiO_2_-ratio (P/F-ratio); requirement for cardiopulmonary resuscitation (CPR) before CT; the ISS, including the abbreviated injury severity (AIS) scores for chest, head, face, abdomen, extremities, and external; the TTS, including classifications of the P/F-ratios, rib fractures, contusions, pleural involvement, and age; the polytrauma Berlin definition, including classifications of injuries with AIS scores of ≥3 in ≥2 body regions (2AIS ≥ 3), combined with the presence of ≥1 physiological risk factors among systolic blood pressure (SBP) ≤90 mmHg, coagulopathy (partial thromboplastin time (PTT) ≥40 s or international normalized ratio (INR) ≥1.4), acidosis (base excess, BE, ≤−6.0 mmol/L)), a Glasgow coma scale (GCS) score ≤ 8 points, and age ≥ 70 years. The tube-to-carina distances were measured on CT to identify patients with deep tracheal tube positions (0–2 cm) and endobronchial malposition (<0 cm). The investigated outcome factors were length of stay in the intensive care unit (ICU), ventilator days, 24-h mortality, 30-day mortality, and hospital mortality. 

To identify possible predictive associations with chest tube placement that were independent from the ISS or TTS (which were, in part, based on CT findings), multivariable logistic regression analysis was performed, including significant associations of univariable analysis. The alpha level of significance was set at 0.05. Receiver operating characteristic (ROC) curve analysis was performed to assess the diagnostic accuracy of the CT-dependent trauma scores with chest tube placement. The area under the ROC curve (AUC) results were considered failed for AUC values between 0.5–0.6, poor for AUC values between 0.6–0.7, moderate for AUC values between 0.7–0.8, good for AUC values between 0.8–0.9, and excellent for AUC values between 0.9–1.

In another step, cases of chest tube placement without distinctive chest injury were excluded to assure that only patients with reproducible indications for chest tube placement were analyzed. Sample size estimation was not performed due to the exploratory and retrospective study approach with mainly descriptive characteristics. The analysis was performed using R (R Foundation for Statistical Computing, Vienna, Austria. https://www.R-project.org, accessed on 6 February 2022) and GraphPad Prism version 9.3.1 for Windows, GraphPad Software, San Diego, CA, USA.

## 3. Results

### 3.1. Demographic Data

During the study period, 5542 trauma team activations occurred, including 616 patients who fully met the inclusion criteria and underwent further analysis (Figure 1 and Table 1). Most of the patients were male (76%), the median age was 50 years (IQR 34), the median ISS was 26 (IQR 21), and the median TTS was 6 (IQR 7). Fifty percent of the patients fulfilled the polytrauma Berlin definition criteria, and the hospital mortality was 24.9%. Blunt trauma mechanisms accounted for 97% of the injuries (51% from road traffic accidents, 37% from falls, and 9% from other blunt trauma mechanisms), and 3% of the injuries were caused by penetrating trauma mechanisms. CPR before CT was performed in 78 patients (12.7%), chest tube placement was performed in 224 patients (36.3%), and thoracotomy was performed in 10 patients (1.6%).

### 3.2. Characteristics of Patients Undergoing Chest Tube Placement

The patients undergoing chest tube placement were significantly younger (*p* = 0.005), had higher ISS and TTS scores (*p* < 0.001 and *p* < 0.001, respectively), met the Berlin polytrauma definition criteria in greater proportions (*p* < 0.001), had lower AIS head (*p* < 0.001), had poorer P/F-ratios (*p* < 0.001), underwent CPR before CT more frequently (*p* < 0.001), had more ventilator days and longer lengths of stay at the ICU (*p* = 0.006 and *p* = 0.008, respectively), and had higher 24-h mortality rates (*p* = 0.022) than patients who did not receive chest tube placement. The sex, BMI, 30-days mortality, and hospital mortality of the groups were comparable (Table 1). 

### 3.3. Chest Tube Placement in Relation to Initial Computed Tomography

#### 3.3.1. Chest Tube Placement before CT

Chest tube placement was performed before CT in 142 patients (23%). In 51 of these patients (35.9%), chest tube placement was performed by prehospital EMS, and in 105 patients (73.9%), in the trauma room of the ED (including 14 patients who had already undergone prehospital chest tube placement) (Table 2). 

Of these 142 patients, 125 (88%) showed significant chest injury on CT (e.g., rib fractures or lung contusions) and 17 (12%) did not (Table 3). In total, 10 of these 17 patients had undergone correction of the tracheal tube position before CT, at least six of which were performed due to transient endobronchial mainstem intubation according to the charts. Four patients had unrecognized endobronchial mainstem intubation confirmed on CT that was corrected after CT (including two with tube correction before CT), and five had unknown reasons for chest tube placement. 

Patients receiving chest tube placement in the ED tended to be more severely injured (with a higher ISS, TTS, and Berlin definition proportion, and lower P/F-ratio) and had higher mortality rates than patients with prehospital chest tube placement, although detailed comparisons were not possible because of 14 patients who received chest tube placement in both locations, who presented with the poorest conditions (Table 2). 

#### 3.3.2. Chest Tube Placement after CT

Of the 474 patients who did not receive chest tube placement before CT, 82 (17.3%) underwent chest tube placement after CT (Table 4). Of these, 56 patients (68.3%) had a clinically relevant pneumothorax, whereas 26 patients (31.7%) had a small pneumothorax.

Of the remaining 392 patients without chest tube placement, five (1.2%) had a small pneumothorax on CT without chest tube placement in due course.

A total of 60 patients (42%) who had undergone chest tube placement before CT received another chest tube placement after CT, of which, 15 patients (25%) had distinctive indications on CT (large pneumothorax at the contralateral chest side) and 45 (75%) had functionality issues or malposition requiring replacement (Table 4). Of the 82 patients who did not receive another chest tube placement after CT, four (4.8%) had small pneumothorax at the contralateral chest side that were monitored without the need for chest tube placement. 

Patients with chest tube placement before and after CT presented with significantly lower P/F-ratios and higher TTS than patients with chest tube placement before CT and not after CT, and patients with chest tube placement after CT and not before CT (P/F-ratio, *p* = 0.041, and *p* = 0.005, respectively, and TTS, *p* = 0.003, and *p* = 0.002, respectively). The 30-day mortality rates were comparable. The small subgroup of eight patients receiving chest tube placement at three locations (at prehospital EMS, in the trauma room of the ED, and after CT) appeared with the poorest conditions.

### 3.4. Factors Associated with Chest Tube Placement 

Univariable associations independent from the CT findings (excluding the ISS and TTS as reflecting the CT findings) were found for age, an SBP < 90, coagulopathy, acidosis, CPR before CT, and the P/F-ratio (Table 5). After multivariable logistic regression analysis including the five most significant variables (the P/F-ratio, an SBP ≤ 90, coagulopathy, acidosis, and CPR before CT), independent associations for patients requiring chest tube placement were found for the P/F-ratio (OR 0.99; 95% CI 0.99–0.99; *p* < 0.001) and acidosis (OR 5.31; OR 3.65–7.74; *p* = 0.001). 

ROC curve analysis of the trauma score-based associations of chest tube placement presented moderate for the ISS (AUC 0.759; 95% CI 0.718–0.800), good for the TTS (AUC 0.854; 95% CI 0.823–0.885) and AIS chest (AUC 0.875; 95% CI 0.849–0.900), and excellent for TTS pleural involvement (AUC 0.939; 95% CI 0.919–0.960) (Figure 2).

After the exclusion of 43 patients (17 without significant chest injury with chest tubes placed before CT, and 26 presenting with small pneumothorax on CT; Table 3 and Table 5), there remained 181 patients (80.8% of 224 patients) with clearly reproducible indications for chest tube placement on CT. The results of the multivariable logistic regression analysis were confirmed in the new dataset presenting with similar effect sizes for the P/F-ratio (OR 0.99; 95% CI 0.99–0.99; *p* < 0.001) and acidosis (OR 5.61; 95% CI 3.83–8.22; *p* = 0.001). ROC curve analysis of the trauma score-based associations of chest tube placement in this subgroup presented moderate for the ISS (AUC 0.749; 0.705–0.793), good for the TTS (AUC 0.884; 0.855–0.913) and AIS chest (AUC 0.888; 0.864–0.911), and excellent for TTS pleural involvement (AUC 0.947; 0.929–0.966).

## 4. Discussion

Our results show that one third of the mechanically ventilated trauma patients who received initial trauma-CT imaging underwent chest tube placement. Of these, two thirds received chest tube placement before CT imaging. A radiologically comprehensible indication for chest tube placement was confirmed in the majority of these patients. An equal proportion of patients had chest tube placement after CT imaging. The majority of these cases had a relevant pneumothorax, whereas in patients who had already undergone chest tube placement before CT, new chest tube placements were mostly performed to address functionality issues or to correct malposition.

One obstacle for the decision-making for chest tube placement before the initial chest CT is that the presence of relevant pneumothorax or hemothorax may not always be discovered clinically on-site. Furthermore, the duality of the thoracic caves creates the possibility of there being true-positive and false-negative indications at the same time. This is particularly relevant in the time-sensitive circumstances of trauma resuscitation. In unstable patients or under trauma-related CPR, bilateral mini-thoracotomy and chest tube placement should always be performed to rule out life-threatening tension pneumothorax [4,5,6,7]. However, there are patients who still compensate for critical conditions, but are at considerable risk for rapid deterioration if they remain untreated. Identifying these patients and reliably discovering pneumothorax in the prehospital setting or the trauma room is difficult and not always possible by clinical means, and pleural decompression is often performed according to auscultation and/or under the consideration of the underlying chest trauma mechanisms. Trauma patients without significant clinical signs of pneumothorax who are treated by a trauma team with a restrictive approach regarding chest tube placement are at risk of being undertreated. Conversely, patients treated by a trauma team with a more liberal decision-making for chest tube placement may be overtreated and may unnecessarily receive chest tubes. 

Point-of-care ultrasound (POCUS) devices in the prehospital and trauma room settings for the confirmation of pneumothorax prior to CT might contribute to avoiding unnecessary chest tube placement until CT diagnostics [14,15,16,17]. However, it should be noted that the interpretation of ultrasound considerably depends on the expertise of the examiner, and thus, CT is currently regarded as the standard of care [18,19,20]. Furthermore, CT images allowing for three-dimensional reconstructions are more accurate for the assessment of the nature and extent of pulmonary injury than a single-view anteroposterior chest radiography [21,22]. This may be particularly important in patients with suspected major chest trauma and tracheal intubation [23,24].

Of the 17 patients with no relevant chest injuries and chest tube placement before CT, more than half of them had issues with deep tracheal tube positions, which might have been the reason for chest tube placement. In a previous study, we found unrecognized endobronchial tube malposition detected on CT in 4.2% of the patients [25]. Seven of them received chest tube placement at the contralateral chest side, four of which were presumably because of the absence of breath sounds. 

Reliable clinical characteristics for identifying the need for chest tube placement are not available for mechanically ventilated trauma patients. In this study, poor P/F-ratios and acidosis indicated patients at risk for chest tube placement. These findings are in line with those of a previously published meta-analysis, in which 91.8% of ventilated patients with traumatic tension pneumothorax presented with hypoxia, in contrast to 50% of patients who were breathing spontaneously [3]. The authors concluded that the reported clinical presentation of tension pneumothorax depends on the ventilatory status of the patient. However, poor P/F-ratios and acidosis are unspecific parameters and, thus, may not be appropriate for identifying patients at risk. Compared with the clinical characteristics, the associations of the trauma scores with chest tube placement were different in our study. Although the ISS showed only moderate association due to several confounding factors (i.e., AIS head), the TTS and AIS chest showed good associations, and pleural involvement showed excellent predictive power, reflecting the findings from the underlying initial CT imaging. 

The management of occult pneumothorax or hemothorax often depends on individual circumstances and is not standardized [26]. High ISS scores, the need for mechanical ventilation, and hemothorax with CT-detected blood collection measuring >1.5 cm were found to increase the likelihood of chest tube placement. The size of the pneumothorax did not appear to be significant in determining the need for chest tube placement [27]. This is in contrast to another study that identified displaced rib fractures and moderate-sized pneumothorax as significant factors associated with chest tube placement in patients with blunt chest trauma and occult pneumothorax [28]. In another study, the conservative treatment of occult hemothorax failed in about 23% of patients, in which hemothorax volumes of more than 300 mL and the need for mechanical ventilation had the highest predictive value for chest tube placement [29]. However, a conservative approach to occult pneumothorax detected only on CT can be a safe procedure, even for mechanically ventilated patients [11,30]. Close cardiopulmonary monitoring is considered important, and appropriate preparations for emergent chest tube placement are essential in case the clinical condition deteriorates [31]. The optimal timing for plain chest radiography, CT follow-up, or bedside pleural ultrasound examination is not provided in the guidelines, but close intervals are recommended in clinically unstable patients [28,32]. 

The main reason for chest tube placement after CT in patients with small pneumothorax in this study cohort was to avoid a potentially increasing pneumothorax under mechanical ventilation. This was particularly relevant in patients with severe lung contusions and impaired gas exchange requiring aggressive mechanical ventilation and high PEEP; patients requiring emergent surgery, particularly craniotomy and laparotomy; or patients with increasing intracranial pressures.

A small number of patients with small pneumothorax did not receive chest tube placement, and were managed conservatively. This reflects the special study cohort including only mechanically ventilated patients.

Regarding patients who received chest tube replacement after CT due to functionality issues or malposition, these patients presented with poorer P/F ratios and higher TTS scores than patients receiving chest tube placement after CT only or before CT only, whereas the 30-day mortality rates were comparable. This is in line with a previous study from our center, in which major trauma patients with malposition of chest tubes on initial CT did not show worse outcomes compared to patients with correctly positioned tubes [33]. 

Hygienic limitations may also be a reason for new chest tube placement after CT because almost all prehospital or CPR-related chest tube placements are considered to be performed not under completely sterile circumstances. 

At every stage of acute emergency care, there is some clinically-based selection for CT and for chest tubes, and the risks and benefits of chest tube placement need to be considered individually. This selection of patients occurs at multiple points along a patient’s trajectory, and comparisons that are not temporally synchronized would be subject to selection bias.

### Limitations

We acknowledge the general limitations of retrospective single-center studies. We only included patients who underwent CT following tracheal intubation and mechanical ventilation. Pre-existing pulmonary diseases in patients that could have influenced gas exchange could not be analyzed due to missing data. A considerable number of critical trauma patients who died before CT evaluation, who underwent immediate surgery without CT, who received head CT only, or who were not mechanically ventilated might have presented with different risk factors and outcomes. Although we have increasingly used POCUS for pneumothorax detection in the trauma room in recent years, the findings are not documented systematically and, thus, could not be analyzed.

## 5. Conclusions

In mechanically ventilated trauma patients, the clinical decision for chest tube placement before initial CT imaging may be associated with relevant false-negative and false positive cases. Chest tube placement after CT may be required at frequencies comparable to those for placement before CT, either due to clinically undetected pneumothorax or due to misplaced chest tubes, which underlines the importance of initial CT imaging. These results support the idea that clinical assessment and CT imaging, together, are important indicators for chest tube placement decisions that cannot be provided by using clinical assessments or CT alone.

## Figures and Tables

**Figure 1 jcm-11-04043-f001:**
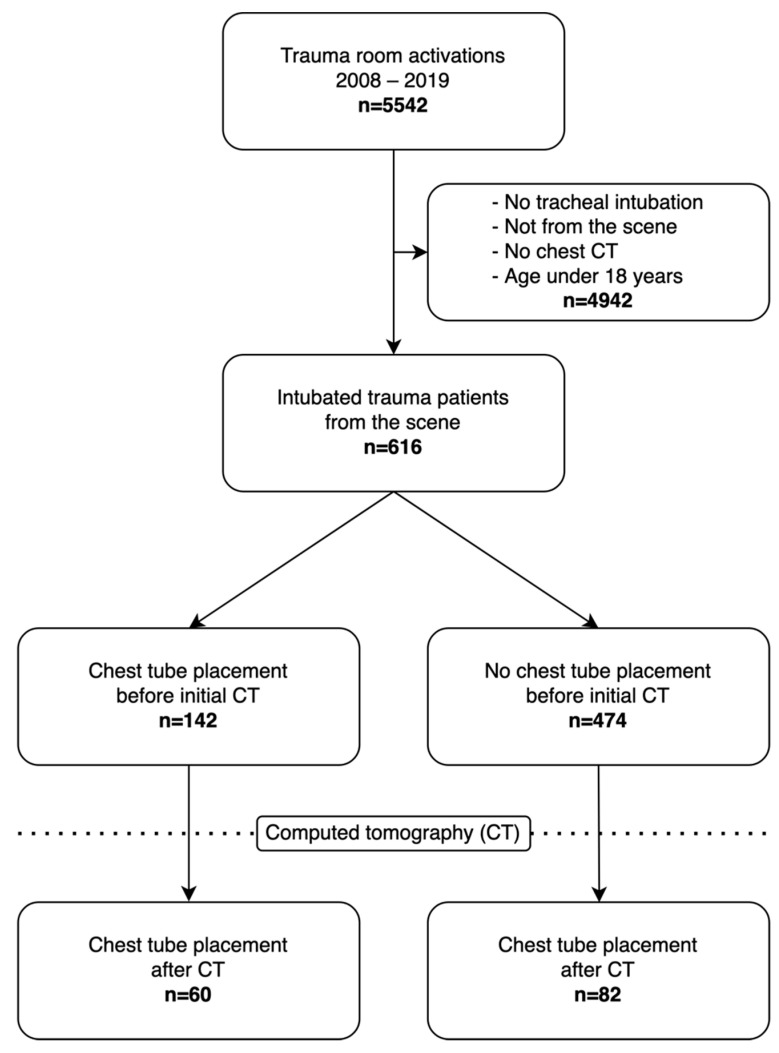
Study flowchart.

**Figure 2 jcm-11-04043-f002:**
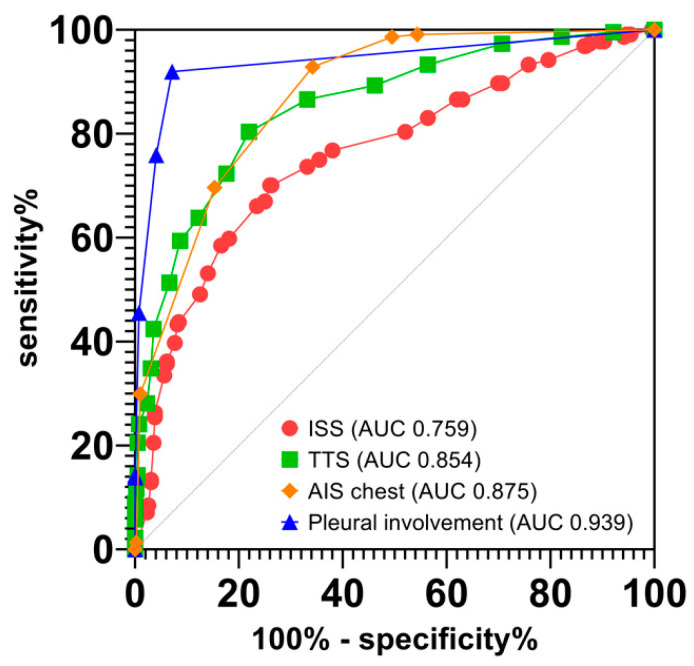
Receiver operating characteristic curve analysis of trauma score associations for chest tube placement. ISS, injury severity score; AUC, area under the receiver operating characteristic curve; TTS, thoracic trauma severity score; AIS, abbreviated injury severity score.

**Table 1 jcm-11-04043-t001:** Baseline characteristics of patients with and without chest tube placement.

	All	Chest Tube Placement	No Chest Tube Placement	*p*-Value
*n*	616	224	392	
Age (years)	50.0 (34.0)	44. (29.0)	53.0 (37.0)	0.005
Male, *n* (%)	448 (72.7)	166 (74.1)	282 (71.9)	0.574
BMI	26.0 (4.0)	26.0 (4.5)	26.0 (4.0)	0.689
ISS	26.0 (21.0)	38.0 (27.5)	25.0 (14.5)	<0.001
AIS chest	3.0 (4.0)	4.0 (2.0)	2.0 (1.8)	<0.001
AIS head	3.0 (5.0)	2.0 (4.0)	2.8 (2.1)	<0.001
AIS face	0.0 (1.0)	0.0 (1.0)	0.5 (0.8)	0.992
AIS abdomen	0.0 (2.0)	0.0 (3.0)	1.0 (1.6)	<0.001
AIS extremity	2.0 (3.0)	2.0 (4.0)	1.5 (1.5)	<0.001
AIS external	0.1 (0.0)	0.0 (0.0)	0.0 (0.0)	0.007
TTS	6.0 (7.0)	11.0 (7.0)	4.0 (4.0)	<0.001
P/F-ratio	1.0 (2.0)	1.0 (3.0)	0.0 (1.0)	<0.001
Rib fracture	0.0 (2.0)	2.0 (3.0)	0.0 (0.0)	<0.001
Contusion	2.0 (2.0)	3.0 (1.0)	0.0 (2.0)	<0.001
Pleural involvement	0.0 (2.0)	2.0 (1.0)	0.0 (0.0)	<0.001
Age	2.0 (2.0)	2.0 (2.0)	2.0 (2.0)	0.007
Berlin, *n* (%)	310 (50.3)	171 (76.3)	139 (35.4)	<0.001
2AIS ≥ 3	320 (51.9)	177 (79.0)	143 (36.4)	<0.001
SBP ≤ 90	172 (27.9)	105 (46.8)	67 (17.0)	<0.001
Coagulopathy	122 (19.8)	80 (35.7)	42 (10.7)	<0.001
Acidosis	174 (28.2)	111 (49.5)	63 (16.0)	<0.001
Age ≥ 70	137 (22.2)	43 (19.2)	103 (41.4)	0.049
GCS ≤ 8	493 (80.0)	141 (62.9)	262 (66.8)	0.334
CPR prior CT, *n* (%)	78 (12.7)	44 (13.4)	34 (8.6)	<0.001
P/F-ratio	393 (161.5)	321 (257)	410 (87.5)	<0.001
Ventilator (days)	3.0 (12.7)	4.5 (17.0)	3.0 (16.0)	0.006
ICU (days)	8.0 (20.0)	10.0 (25.0)	7.0 (19.0)	0.008
24-h mortality, *n* (%)	59 (9.6)	30 (13.4)	29 (7.2)	0.022
30-d mortality, *n* (%)	147 (23.9)	59 (26.3)	88 (22.4)	0.281
Hospital mortality, *n* (%)	153 (24.8)	64 (28.5)	89 (22.7)	0.120

BMI, body mass index; ISS, injury severity score; AIS, abbreviated injury severity; TTS, thoracic trauma score; P/F-ratio, PaO_2_/FiO_2_-ratio; Berlin, Berlin polytrauma definition; SBP ≤ 90, systolic blood pressure < 90 mmHg; Age ≥ 70, age ≥ 70 years; GCS ≤ 8, Glasgow coma scale ≤ 8 points; CPR, cardiopulmonary resuscitation; CT, computed tomography; ICU, intensive care unit.

**Table 2 jcm-11-04043-t002:** Characteristics of patients with chest tube placement before initial computed tomography.

	Chest Tube before CT	Chest Tube EMS	Chest Tube ED	Chest Tube EMS+ED	Chest Tube before CT Only
*n*	142	51	105	14	82
Age (years)	47.5 (28.0)	48.0 (27.0)	46.0 (28.0)	49.0 (26.0)	48.0 (28.0)
Male, *n* (%)	112 (78.8)	46 (90.2)	78 (74.2)	12 (85.7)	65 (79.2)
BMI	26.0 (4.0)	25.0 (4.0)	26.0 (3.0)	26.0 (3.0)	26.0 (3.0)
ISS	37.0 (29.0)	34.0 (21.0)	43.0 (28.0)	42.0 (28.0)	34.0 (26.0)
AIS chest	4.0 (2.0)	4.0 (2.0)	4.0 (2.0)	4.5 (1.0)	4.0 (1.0)
AIS head	2.0 (5.0)	0.0 (4.0)	2.0 (5.0)	0.0 (4.0)	2.0 (5.0)
AIS face	0.0 (2.0)	0.0 (0.0)	0.0 (2.0)	0.0 (0.0)	0.0 (1.0)
AIS abdomen	0.0 (3.0)	0.0 (3.0)	0.0 (3.0)	0.5 (3.0)	0.0 (2.0)
AIS extremity	2.0 (4.0)	2.0 (3.0)	2.0 (4.0)	2.0 (3.0)	2.0 (3.0)
AIS external	0.0 (0.0)	0.0 (0.0)	0.0 (0.0)	0.0 (0.0)	0.0 (0.0)
TTS	11.0 (9.0)	11.0 (8.0)	11.0 (8.5)	16.0 (8.0)	10.0 (6.0)
P/F-ratio	2.0 (5.0)	1.0 (5.0)	2.0 (4.0)	5.0 (3.0)	1.0 (5.0)
Rib fracture	2.0 (3.0)	2.0 (3.0)	2.0 (2.0)	3.0 (1.0)	2.0 (3.0)
Contusion	3.0 (1.0)	3.0 (1.0)	3.0 (1.0)	3.0 (2.0)	2.0 (1.0)
Pleural involvement	2.0 (1.0)	2.0 (2.0)	2.0 (1.0)	3.0 (2.0)	2.0 (2.0)
Age	2.0 (2.0)	2.0 (2.0)	2.0 (1.0)	2.0 (2.0)	2.0 (2.0)
Berlin, *n* (%)	107 (75.3)	38 (62.7)	82 (78.1)	13 (92.8)	56 (68.3)
2 AIS ≥ 3	109 (76.7)	38 (62.7)	84 (80.0)	13 (92.8)	57 (69.5)
SBP ≤ 90	71 (50.0)	21 (41.2)	21 (41.2)	10 (71.4)	37 (45.1)
Coagulopathy	58 (40.8)	19 (37.2)	48 (45.7)	9 (64.3)	31 (37.8)
Acidosis	71 (50.0)	27 (52.9)	56 (53.3)	12 (85.7)	34 (41.4)
Age ≥ 70	21 (14.7)	8 (15.7)	14 (13.3)	1 (7.1)	12 (14.6)
GCS ≤ 8	88 (62.0)	30 (58.8)	68 (64.7)	10 (71.4)	50 (61.0)
CPR prior CT, *n* (%)	35 (24.6)	11 (21.5)	30 (28.6)	6 (32.8)	19 (23.2)
P/F-ratio	280 (283)	346 (291)	223 (290)	134 (185)	362.5 (275)
Ventilator (days)	6.0 (12.5)	6.0 (12.5)	6 (11.0)	8.0 (16.0)	4.5 (16.0)
ICU (days)	9.5 (21.0)	8.0 (19.0)	10 (24.0)	10.5 (19.0)	8.0 (19.0)
24-h mortality, *n* (%)	27 (19.0)	9 (17.6)	22 (20.9)	4 (28.6)	18 (21.9)
30-d mortality, *n* (%)	45 (31.6)	14 (27.4)	38 (36.2)	7 (50.0)	27 (32.9)
Hospital mortality, *n* (%)	50 (35.2)	18 (35.3)	41 (39.0)	7 (50.0)	30 (36.6)
Chest tube after CT	60 (42.2)	18 (35.3)	50 (47.6)	8 (57.1)	N/A

BMI, body mass index; ISS, injury severity score; AIS, abbreviated injury severity; TTS, thoracic trauma score; P/F-ratio, PaO_2_/FiO_2_-ratio; Berlin, Berlin polytrauma definition; SBP ≤ 90, systolic blood pressure ≤ 90 mmHg; Age ≥ 70, age ≥ 70 years; GCS ≤ 8, Glasgow coma scale ≤ 8 points; CPR, cardiopulmonary resuscitation; CT, computed tomography; ICU, intensive care unit; N/A, not applicable.

**Table 3 jcm-11-04043-t003:** Individual characteristics of patients with chest tube placement before computed tomography and minor thoracic injuries.

Case	Age (Years)	Gender	ISS	AIS Chest	AIS Head	TTS	P/F	CPR before CT	ETT Correction before CT	TCD on CT (cm)	Chest Tube EMS	Chest Tube ED	Chest Tube after CT	Outcome
1	94	male	29	2	1	7	428	No	Yes	7.3	-	R	-	Survived
2	36	male	22	2	1	4	361	No	No	−0.5	-	L	-	Survived
3	40	male	18	1	3	4	418	No	No	6.7	R	-	-	Survived
4	48	male	24	0	5	3	423	No	Yes	6.5	L	-	-	Survived
5	73	female	36	2	4	3	377	Yes	Yes	−1.3	-	L	-	Died day 1
6	23	male	18	0	2	2	512	No	No	6.3	R	-	-	Survived
7	48	male	33	2	3	4	78	Yes	Yes	−0.5	-	L	L	Died day 11
8	34	male	32	2	3	4	423	No	Yes	4.7	-	L	-	Survived
9	37	female	43	2	3	6	496	No	Yes	3.2	-	L	L	Survived
10	37	male	4	0	0	3	517	No	Yes	2.3	R+L	-	-	Survived
11	24	male	4	0	0	0	586	No	No	5.4	R	-	-	Survived
12	19	male	8	1	0	3	455	No	No	2.9	-	R	-	Survived
13	18	female	75	2	6	6	29	Yes	No	−1.5	-	R+L	-	Died day 1
14	25	male	29	1	2	4	446	No	Yes	5.1	-	L	-	Survived
15	23	male	34	2	3	6	160	No	Yes	5.3	-	R+L	-	Survived
16	49	female	75	2	6	7	218	No	Yes	5.0	-	L	-	Died day 71
17	27	male	24	1	4	2	259	No	No	7.8	-	R	-	Survived

ISS, injury severity score; AIS, abbreviated injury severity score; TTS, thoracic trauma severity score; P/F, PaO_2_/FiO_2_-ratio; CPR, cardiopulmonary resuscitation; CT, computed tomography; ETT, endotracheal tube; TCD, tube-to-carina distance; EMS, emergency medical service; ED, emergency department; R, right-sided; L, left-sided.

**Table 4 jcm-11-04043-t004:** Characteristics of patients with chest tube placement after initial computed tomography.

	Chest Tube after CT	Chest Tube after and Not before CT	Chest Tube after and before CT	Chest Tube EMS, ED, and after CT
*n*	142	82	60	8
Age (years)	44.0 (29.0)	42.0 (33.0)	46.5 (27.5)	49.0 (24.5)
Male, *n* (%)	101 (71.1)	54 (65.8)	47 (78.3)	7 (87.5)
BMI	25.0 (5.0)	25.0 (5.0)	26.0 (4.0)	26.0 (5.5)
ISS	41.0 (22.0)	41.0 (22.0)	42.0 (22.5)	37.5 (17)
AIS chest	4.0 (1.0)	4.0 (2.0)	4.0 (1.0)	4.0 (1.0)
AIS head	3.0 (4.0)	3.0 (4.0)	2.0 (5.0)	0.0 (3.5)
AIS face	0.0 (2.0)	0.0 (1.0)	0.0 (2.0)	0.0 (0.0)
AIS abdomen	2.0 (3.0)	2.0 (3.0)	2.0 (5.0)	2.5 (3.0)
AIS extremity	3.0 (2.0)	3.0 (4.0)	3.0 (2.0)	3.0 (1.5)
AIS external	0.0 (0.0)	0.0 (0.0)	0.0 (0.0)	0.0 (0.0)
TTS	11.0 (7.0)	10.0 (7.0)	12.0 (8.0)	12.5 (9.5)
P/F-ratio	1.0 (2.0)	1.0 (2.0)	2.5 (4.0)	3.0 (4.0)
Rib fracture	2.0 (2.0)	2.0 (3.0)	2.0 (2.0)	2.5 (2.0)
Contusion	2.0 (3.0)	3.0 (1.0)	3.0 (1.0)	3.0 (0.5)
Pleural involvement	3.0 (1.0)	2.0 (2.0)	3.0 (1.0)	3.0 (2.0)
Age	3.0 (1.0)	1.5 (3.0)	2.0 (2.0)	2.0 (2.0)
Berlin, *n* (%)	115 (81.0)	64 (78.0)	51 (85.0)	7 (87.5)
2 AIS ≥ 3	120 (84.5)	68 (82.9)	52 (86.6)	7 (87.5)
SBP ≤ 90	68 (47.9)	34 (41.4)	34 (56.6)	5 (62.5)
Coagulopathy	49 (34.5)	22 (26.8)	27 (45.0)	4 (50)
Acidosis	79 (75.6)	42 (51.2)	37 (61.6)	6 (75)
Age ≥ 70	22 (15.5)	13 (15.8)	9 (15.0)	1 (12.5)
GCS ≤ 8	91 (64.1)	53 (64.6)	38 (63.3)	7 (87.5)
CPR prior CT, *n* (%)	25 (17.6)	9 (10.9)	16 (26.6)	2 (25)
P/F-ratio	283.0 (236.0)	334.5 (202.0)	206.5 (281.5)	186.0 (238.5)
Ventilator (days)	10.0 (17.0)	11.0 (16.0)	8.5 (16.0)	15 (20.75)
ICU (days)	15.0 (23.0)	17.0 (22.0)	13.0 (24.0)	17.0 (24.0)
24-h mortality, *n* (%)	12 (8.4)	3 (3.6)	9 (16.0)	1 (12.5)
30-d mortality, *n* (%)	32 (22.5)	14 (17.1)	18 (30.0)	2 (25.0)
Hospital mortality, *n* (%)	34 (23.0)	14 (17.1)	20 (33.3)	2 (25.0)

BMI, body mass index; ISS, injury severity score; AIS, abbreviated injury severity; TTS, thoracic trauma score; P/F-ratio, PaO_2_/FiO_2_-ratio; Berlin, Berlin polytrauma definition; SBP ≤ 90, systolic blood pressure ≤ 90 mmHg; Age ≥ 70, age ≥ 70 years; GCS ≤ 8, Glasgow coma scale ≤ 8 points; CPR, cardiopulmonary resuscitation; CT, computed tomography; ICU, intensive care unit.

**Table 5 jcm-11-04043-t005:** Associations of clinical characteristics with chest tube placement.

	Univariable or (95% CI)	*p*-Value	Multivariable or (95% CI)	*p*-Value
Age	0.98 (0.97–0.99)	0.003		
Male	1.11 (0.77–1.61)	0.561		
BMI	1.02 (0.97–1.06)	0.357		
SBP ≤ 90	4.28 (1.95–6.20)	<0.001	1.58 (0.93–2.67)	0.088
Coagulopathy	4.63 (3.04–7.05)	<0.001	0.91 (0.46–1.78)	0.782
Acidosis	5.31 (3.65–7.74)	<0.001	2.54 (1.48–4.37)	0.001
Age ≥ 70	0.50 (0.33–0.77)	0.001		
GCS ≤ 8	0.84 (0.59–1.18)	0.329		
CPR before CT	2.57 (1.59–4.17)	<0.001	0.61 (0.32–1.16)	0.131
P/F-ratio	0.99 (0.99–0.99)	<0.001	0.99 (0.99–0.99)	<0.001

BMI, body mass index; SBP ≤ 90, systolic blood pressure ≤ 90 mmHg; GCS ≤ 8, Glasgow coma scale ≤ 8 points; CPR, cardiopulmonary resuscitation; CT, computed tomography; P/F-ratio, PaO_2_/FiO_2_-ratio; Covariates of the multivariable analysis included the most significant variables of the univariable analysis (*p* < 0.001).

## Data Availability

The data supporting the findings of this study are available from the corresponding author upon reasonable request.

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
