# Peer review of "Chest Tube Placement in Mechanically Ventilated Trauma Patients: Differences between Computed Tomography-Based Indication and Clinical Decision"

_jcm, 2022, doi:10.3390/jcm11144043_

Round 1

Reviewer 1 Report

Thank you for this. I found this interesting from a respiratory perspective, as i am a physician but we look after trauma locally. our practice would mirror the findings, and it is good to see this in writing. i think the methodology is sound, limitations spoken of and results presented in a reasonable fashion. I like the discussion of conservative vs interventional management.

Author Response

RESPONSE: We would like to thank the Reviewer for his efforts and appreciate his positive evaluation.

Reviewer 2 Report

The research question about the differences between computed tomography-based indication and clinical decisions for chest tube placement in mechanically ventilated patients is interesting. There is not much literature available regarding the topic. The relevance of the study may be of limited use due to the increasing use of ultrasound - this is, however, acknowledged by the authors in the manuscript.  The manuscript is very well written. The text is clear, easy to read, and without any obvious grammatical erros.  The research methodology and the statistical analysis used are appropriate for the data collected. The results and conclusions drawn from the presented evidence seem accurate.  The researchers were able to answer their initial question and concluded that both clinical assessment and imaging should be utilized for decision making. It will be a good addition to literature as while this approach is used in clinical practice, there is a dearth of data to support it and this manuscript helps to overcome some of those gaps.

Author Response

RESPONSE: We would like to thank the Reviewer for his efforts and are thankful for his positive response.

Reviewer 3 Report

This is a very interesting study which gives the proof of needing both radiological and clinical approach and not only one of the above.

I have some comments:

- Was the sample size estimated? Please specify in data analysis section.

- Although not being the aim of this study, it would be interesting to add some words in discussion about a comparison between CT scan and classic chest x-ray regarding chest tube placement.

-Please add data about pre-existing pulmonary diseases among the study population and analyze it. It would be really notable to see if this affects your results.  

- I found some English errors throughout the paper. Please have a re-check.   

Author Response

This is a very interesting study which gives the proof of needing both radiological and clinical approach and not only one of the above.

I have some comments:

- Was the sample size estimated? Please specify in data analysis section.

RESPONSE: We would like to thank the reviewer for this suggestion.  We did not perform a formal sample size calculation - instead, the available patients served as the sample size for this study.

We have added to the methods part: “Sample size estimation was not performed due to the exploratory and retrospective study approach with mainly descriptive characteristics.”

Nonetheless, due to the reviewer’s suggestion, we have performed a sample size estimation in groups of patients with and without chest tube placement according to the exact effect size classification by Cohen considering conventional effect size values of 0.2 small, 0.5 medium, and 0.8 large at a power of greater than 80%.

Given our group sizes of 224 and 392 patients, we had greater than 80% power to detect a small to medium effect size of 0.23, which can be considered sufficient in the context of the study purpose.

We think that a sample size estimation in a retrospective analysis will not add much to the main message of the paper. If, however, the Editor now insists on the sample size calculation we are fine with the inclusion of these data into the manuscript.

Ref: Cohen, J. (1988). Statistical power analysis for the behavioral sciences (2nd ed.). Hillsdale,NJ: Lawrence Erlbaum.

Below and for transparency reasons, we added the R- code for your convenience:

require(pwr)

## Lade nötiges Paket: pwr

## Warning: Paket 'pwr' wurde unter R Version 4.1.3 erstellt

pwr.2p2n.test(n1 = 224 , n2 = 392, sig.level = 0.05, power = 0.8)

##
##      difference of proportion power calculation for binomial distribution (arcsine transformation)
##
##               h = 0.2346747
##              n1 = 224
##              n2 = 392
##       sig.level = 0.05
##           power = 0.8
##     alternative = two.sided
##
## NOTE: different sample sizes

pwr::cohen.ES(test = "p", size = "small")

##
##      Conventional effect size from Cohen (1982)
##
##            test = p
##            size = small
##     effect.size = 0.2

pwr::cohen.ES(test = "p", size = "medium")

##
##      Conventional effect size from Cohen (1982)
##
##            test = p
##            size = medium
##     effect.size = 0.5

- Although not being the aim of this study, it would be interesting to add some words in discussion about a comparison between CT scan and classic chest x-ray regarding chest tube placement.

RESPONSE: We agree with the Reviewer and have added two more sentences (including four new references) to the discussion section “Furthermore, CT images allowing for three-dimensional reconstructions are more accurate for the assessment of the nature and extent of pulmonary injury than a single-view anteroposterior chest radiography [21,22]. This may particularly be important in patients with suspected major chest trauma and tracheal intubation [23,24].”

-Please add data about pre-existing pulmonary diseases among the study population and analyze it. It would be really notable to see if this affects your results.

RESPONSE: We agree that this would be an interesting point. Unfortunately, pre-existing pulmonary diseases could not be analyzed in this study cohort and are no longer available due to the retrospective and anonymized design. However, this would be a desirable study approach of a future investigation. The analysis of a potential influence of pre-existing diseases should then include clinical presentation before and after tracheal intubation (SpO2), adjustment on mechanical ventilation details (PIP, PEEP, FiO2) and assessment of oxygenation, CO2 elimination, compliance and resistance. For now, we have added the following phrase to the limitations section: “Pre-existing pulmonary diseases of patients which could have influenced gas exchange could not be analyzed due to missing data.”

- I found some English errors throughout the paper. Please have a re-check.   

RESPONSE: Accordingly, we have checked language and grammar again by a professional processing service (MDPI: English editing invoice:english-46815).

We would like to thank the reviewer for his valuable suggestions. We really appreciate his efforts and feel that the manuscript has now considerably improved.

Thank you again.

Round 2

Reviewer 3 Report

Authors replied to my comments satisfactorily. IMHO this manuscript can be accepted now.